# OpenReview forum: "Bias Runs Deep: Implicit Reasoning Biases in Persona-Assigned LLMs"
_ICLR.cc/2024/Conference — ICLR 2024 poster_

### Official Review · Reviewer_s6hK · 2023-10-28

**Soundness:** 2 fair
**Presentation:** 3 good
**Contribution:** 2 fair
**Rating:** 5
**Confidence:** 4

**Summary:**

This paper evaluates the effect of 16 different persona assignments on ChatGPT’s ability to perform basic reasoning tasks, which span 24 datasets. The authors find large drops in accuracy for certain personas (including physically disabled individuals and religious individuals). One of the main factors identified by the authors for this significant drop in performance was ChatGPT’s tendency to abstain from answering for those personas. The authors also tried a few prompt variants to de-bias the model.

**Strengths:**

Given the increasing number of works interested in using LLMs to simulate human behavior, this work studies an important facet of that application which is the potential bias that is introduced when prompting LLMs to exhibit certain personas. The authors also evaluated ChatGPT across a number of personas and reasoning tasks.

**Weaknesses:**

My primary concern is that this work only evaluates one model, i.e., ChatGPT, which makes it challenging to assess the generalizability of the findings regarding the bias of LLMs. For example, it seems intuitive based on anecdotal evidence that ChatGPT would be more likely to abstain from answering questions for certain personas, but that does not necessarily mean it should be penalized for that behavior. While the authors mentioned in a footnote that they also ran similar experiments on a Llama-2 model (could the authors clarify which model specifically?), it would be helpful to see the same evaluation conducted across a number of different LLMs.

A few comments about providing more experimental details and justification: (1) could the authors clarify how the “human” and “average human” baselines were implemented? (2) Could the authors explain why they chose to sample from ChatGPT with temperature 0? Relatedly, since it was mentioned that there was significant variation in the model’s performance across different runs, could the authors include error bars in the results analysis?

The approaches considered to de-bias LLMs seemed to be very handcrafted. Could the authors try a more systematic approach to exploring the prompt space or types of examples that could be provided to the LLMs to de-bias the model? It would be interesting to see de-biasing results across more LLMs.

---
I have raised my score accordingly as a result of the additional experiments on LLMs beyond ChatGPT. I am still concerned about the justification of the baselines, the author's choice of de-biasing approaches and am unsure about their implications on future methodology.

**Questions:**

- Please clarify the questions raised in the weakness section above.
- It would be helpful for informing future work and for gaining a deeper understanding of the findings in this work to be able to understand *why* certain personas experience a larger performance drop. What are systematic approaches to getting at this question?
- I also wonder if these biases are more prevalent in objective questions (where there is a ground truth answer) as compared to subjective questions or tasks.

---

> ### Author Response · Authors · 2023-11-23
> **Response from the authors to Reviewer s6hK**
>
> Thank you for recognizing the importance of this work. Please see our responses below to your specific questions and concerns.
>
> ---
> **Evaluation with other LLMs**
>
> Thanks for the suggestion to incorporate additional LLMs. We have expanded our analysis to include 3 additional models: 2 latest OpenAI models (GPT-4-Turbo and GPT3.5-Turbo Nov.) and 1 public model (Llama2-70b-chat). Upon evaluation across all 24 of our datasets, we found our findings to generalize to these models and the persona-induced biases to be prevalent across models (Figs. 12, 16, and 18). Thus, it is clearly not an issue with just ChatGPT or the OpenAI family of models, but is more fundamental to how the LLMs are currently developed. Please refer to the general response and newly added Appendix D for additional results and analysis.
>
> ---
> **Should Abstentions be penalized?**
>
> We want to clarify that ChatGPT primarily abstains by citing various incorrect (_a physically disabled person would not be able to analyze legal situations_) and stereotypical (_a religious person does not possess any expertise in physics_) presumptions about the socio-demographic personas it is emulating. The abstention examples in Table 3 and Appendix H demonstrate this troubling trend and clarify this key distinction from regular model refusals (e.g. _”I will not provide a bomb making recipe”_). This undesirable behavior would be detrimental to specific groups, e.g. personalized assistants (LLMs with assigned personas aligned with the end-user) responding incorrectly or inappropriately.
>
> ---
> **Better exploration of the prompt space for bias mitigation?**
>
> We explored prompts that go from providing a nudge (“Try your best”) to strongly suggesting to answer a question (“Don’t refuse”) to even making each persona a subject expert. We have extended this set to include results for 8 additional de-biasing prompts (see the general response above). Our mitigation experiments now cover 13 debiasing prompts to more systematically cover the space of variations. We did not see any notable differences in mitigation in the results. Note that the focus of this work is to identify and characterize the reasoning biases in persona-assigned LLMs, and we are the first ones to do so to the best of our knowledge. We leave more sophisticated bias mitigation approaches such as aligning LLMs on our dataset of 1.5 Million released model predictions to future work.
>
> ---
> **Why do certain personas experience a larger performance drop?**
>
> This is a very interesting question. We believe that many factors are at play here. Firstly, modern language models go through multiple training stages like pretraining, instruction-tuning, alignment, etc. Biases within the datasets used at each stage can favor or disfavor certain socio-demographic groups (e.g. text snippets like ‘religious individuals lack interest in sciences,’ or ‘Jewish individuals possess high intelligence’), consequently shaping the induced biases in these models. Secondly, the alignment efforts guided by [red-teaming](https://huggingface.co/blog/red-teaming) for these LLMs might primarily target widely recognized harms and biases, potentially leaving other biases unaddressed, in turn contributing to some personas exhibiting larger performance drops.
>
> Using open pre-training corpuses (e.g. C4, Dolma) one could in principle run ablations on the pre-training corpus to identify why certain personas have a much higher performance drop. E.g. If pre-training on a corpus excluding certain subreddits results in reducing the bias, we can identify that subreddit as a potential cause of this drop. However, in practice such ablations would be prohibitively expensive to run.
>
> We will add this to the discussion section in the final version of the paper.
>
> ---
> **How are the “human” and “average human” baselines implemented?**
>
> To create these baselines, we prompt the LLM to adopt the persona of a “Human” and “Average Human” respectively and measure the resulting performance on our 24 dataset benchmark. Similar to the personas studied in this work, we simply replace the “{persona}” placeholder in persona instructions with “Human” or “Average Human”. We have clarified this in Section 3.1 of the updated paper (see the changes in blue).
>
> ---
> **Why sample from ChatGPT with temperature 0?**
>
> We use a temperature of 0 as it helps us quantify and characterize the bias from the most likely (and least stochastic) responses from these models.
>
> ---
> **Error bars and variation across model runs**
>
> Thank you for pointing this out. We have added Fig. 23 to Appendix F that shows the variation in performance across runs. We found that the standard deviation of the performance across runs (when aggregated over all 24 datasets) is very small (less than 0.006 and is thus barely visible in the figure) and thus doesn’t affect our findings. We have updated the paper to drop the “significant” terminology when discussing the variation across runs.

---

### Official Review · Reviewer_mMGg · 2023-10-30

**Soundness:** 3 good
**Presentation:** 3 good
**Contribution:** 3 good
**Rating:** 5
**Confidence:** 4

**Summary:**

LLMs can ascribe to a persona by being prompted to respond as a person with specific characteristics. This paper examines biases revealed by persona-assignment that may otherwise go undetected. The results show that being prompted to take on a particular persona reveals stereotypical responses compared to being prompted to comment on that persona directly. The paper mainly includes an experimental demonstration of this problem.

**Strengths:**

(1) The paper underlines a distinction between persona-assigned stereotype bias and more general word association bias. This is an important result in the discussion about bias in LLMs

(2) Section 4 of the analysis is quite interesting where the model abstains from responding for various reasons. Similarly Section 5 is interesting. Strengthening these two sections would be a possible path towards improvement.

**Weaknesses:**

(1) The related works section is underdeveloped. For instance, the work should be properly situated within the context of existing research on Algorithmic bias. But instead, the paper includes only minimal reference to "bias in models"

(2) The paper is not precise in its definition of bias. From the text, it seems like the paper is mainly interested in stereotypes found in word associations. Using 'bias' as a general all encompassing term weakens the overall point of the paper. It would be helpful to be more precise.

(3) Results overstated. The paper states "In summary, these results indicate that there is significant bias inherent in the LLM for nearly all personas, both at the group level (e.g., religion-based personas) as well as for individual personas." Figure 2 shows that for the "Race" categories, the accuracy ranges from ~60% to ~63% compared to ~65% for human persona. With the accuracy being so low across the board, this does not seem to be a "significant" bias. The two bars between man and woman are almost indistinguishable and also very close to human persona. The two groups with differences seem to be physically disable and religious where the accuracy is worse than a coin toss. The paper does not include discussion in the context of such low accuracy. This continues where throughout results section where the paper makes claims like "nearly all personas have large drops on some datasets" which is not a particularly well-supported or meaningful claim.

(4) Analysis is not statistically sound. For instance, Figure 3 "using a scatter plot with various personas on the x-axis. Each point now corresponds to a relative drop in accuracy on one dataset, compared to the Human baseline. The box represents the 25th -75th percentile...". The y-axis is "relative % change". The paper does not outline assumptions being made here about direct comparison between datasets. Further, the accuracy reported does not include error bars so how can you justify comparison in this way?

(5) Figure 8 switches from 'Accuracy' to error rate with no supported discussion. The plot is also scaled from 0.0 to 0.6 without explanation. The general discussion is overstated without appropriate experimental and analytical details to support.

**Questions:**

(1) This work could be viewed as simply a evidence against asking an LLM to take on a human persona. As in, it is not just that assigning personas can surface "deep-rooted biases" but possibly that the act of asking an LLM to take on a human persona is potentially flawed in some fundamental way. What do the authors mean by "deep-rooted biases" and "detrimental side-effects"? Can you be more specific in terms of the connecting the results in this paper to some larger take away?


(2) In table 2, the "Race" category contains race (caucasian), ethnicity (hispanic) and nationality (African). Can you explain this choice? Why "African" person instead of "African American" for instance? And given that you have the flexibility to do so, what is the reason for not including a non-binary gender option?

(3) Why is the y-axis in Figure 2 not standardized from 0.0 - 1.0 or 0.5 - 1.0? Is this test accuracy? If this is AUC, then the scale should be 0.5 - 1.0. Further, why is the best accuracy ~65% across the board? Might this limit the significance of the results?

---

> ### Author Response · Authors · 2023-11-23
> **Response from the authors to Reviewer mMGg [Part 1 of 2]**
>
> Thank you for taking the time to review our paper! Please see our responses below.
> &nbsp;
>
> ---
> ## Significance of the results
> &nbsp;
>
> **Significant bias for nearly all personas**
>
> - Figure 2 serves the purpose of demonstrating that there are disparities in performance across personas. Note that this figure plots the accuracy micro-averaged over all 24 of our datasets and thus even small differences in performance can be *statistically* significant. We explicitly mention in the text whenever the differences are not statistically significant (e.g. the note regarding differences not being statistically significant for Caucasian, Man, and Woman personas in Section 3.1).
> - Our claim that _“there is a significant bias inherent in LLMs for nearly all personas”_ can be substantiated through Figure 3 where we show that relative to the “Human” persona, all personas have at least one dataset where there is nearly a 10% drop in the performance. We think a drop of 10% in the model performance just by changing the socio-demographics of the assigned persona is significant and harmful for the unassuming end-users of these personas-assigned LLMs. We will include this clarification in the final version of the paper.
> - We have included an additional plot in Appendix F (**see Fig. 24**) that complements Fig. 2 and 3 and plots the number of datasets with a statistically significant drop in performance for all personas relative to the baseline “Human” persona. We can see in this figure that 16 out of 19 personas have at least one dataset with a statistically significant drop in the performance.
>
> ---
> **Additional clarifications**
>
> - *"accuracy being so low across the board .. doesn’t seem to be significant"*:
> Firstly, note that 65% is not a low score for MMLU tasks using zero-shot prompting. For example [Fu et al ‘23](https://github.com/FranxYao/chain-of-thought-hub/tree/main) reported _5-shot_ scores for gpt-3.5-turbo as 67.3% (compared to our 65% with a zero-shot prompt). Secondly, we run statistical significance tests for all reported findings and clearly state when the differences are not statistically significant.
>
> - *"physically disabled and religious seem to have performance worse than a coin toss."*: Note that the datasets from MMLU have 4 answer choices and thus the coin toss performance would be 25%. None of the personas report an aggregate score below 40%.
>
>
> - *"two bars between man and woman are almost indistinguishable and also very close to human persona"*: This is true, and we explicitly mention in Section 3.1 that the performance differences for Caucasian, Man, and Woman personas are not statistically significant.
>
> &nbsp;
>
> ---
> ## Soundness of the analysis
> &nbsp;
>
> **Figure 3 and dataset-level comparisons**
>
> Figure 3 shows the distribution of dataset-level %change in performance for all personas and primarily serves the purpose of depicting the extent of the bias across personas (_there is no direct comparison between the personas at the dataset granularity in this figure_)  e.g. Physically-Disabled, Religious, and Trump Supporter personas have datasets with 50%+ drop in the performance; nearly all datasets have at least one dataset with a 10% drop; the distribution of %change varies across personas. For dataset-level comparisons, please refer to Fig. 4 and Fig. 5 which provide insights for various persona groups (e.g. Politics, Race), as well as the newly added **Fig. 24** (in App. F) that provides insights for individual personas.
>
> ---
> **Error bars**
>
> We have added **Figure 23** to Appendix F to show the variation in performance across runs. We can see that the standard deviation of the performance across runs (when aggregated over all 24 datasets) is very small (less than 0.006 and is thus barely visible in the figure), and thus the variation across runs doesn’t affect our findings (hence the omission of error bars in the figures). We will add references to Figure 23 and Figure 24 in the final version of the paper.
>
> ---
> **Why switch from 'Accuracy' to error rate in Figure 8?**
>
> Thank you for bringing this up. As mentioned in Section 4, we switch from accuracy to error rate because at this point in the paper we have moved to analyzing the errors and quantifying the contribution of abstentions to the overall error. We have added an additional note about this to the caption of Figure 8 in the updated paper for further clarification (see the change in blue). We hope this clarifies the confusion. We scale the plot down to 0.6 as it is the highest error rate observed across personas.

---

> > ### Author Response · Authors · 2023-11-23
> > **Response from the authors to Reviewer mMGg [Part 2 of 2]**
> >
> > **Definition of bias in this study and related work**
> >
> > Thank you for bringing this up. We believe personified LLMs should provide the same performance on reasoning tasks for different socio-demographic personas and select our personas as such (our normative assumption). When LLMs have statistically significantly different accuracies on the same reasoning tasks for different socio-demographic personas, we refer to this as _bias_. Such bias that causes certain groups to have lower performance has been categorized under representational harms (Barocas et al., 2017; Blodgett et al., 2020) or “Discrimination, Exclusion and Toxicity” harm (Weidinger et al., 2021). E.g. A personalized voice assistant could be given a persona (either by the end-user or by the designer unbeknownst to the user) that mimics the socio-demographic of the end-user. However, this would result in inferior assistance for the entire socio-demographic groups for which we observed a drop in performance in this study. Considering the broad usage of LLM-as-assistants and personification in LLMs ([character.ai](https://beta.character.ai/), [Meta AI Chatbots](https://ai.meta.com/genai/), [Replika](https://replika.com/), [OpenAI Custom GPTs](https://openai.com/blog/introducing-gpts)), we attempt to surface these issues for both LLM designers and LLM users. We are not studying stereotypical word associations but are studying this specific representational harm. We alluded to this in Sec. 6 and have included this discussion in the newly added **Appendix G**. We have similarly updated the related work to elaborate on model biases and harms (see changes to Section 7 marked in blue).
> >
> > ---
> > **Question: Is this work evidence against asking an LLM to take on a human persona? What are detrimental side effects?**
> >
> > Yes, our paper *is* providing evidence against using LLMs to take on a human persona and given the commercial momentum to giving LLMs personas, our finding has significant practical implications for real users. For example, these persona-assigned agents may actively provide incorrect information, exhibit more errors in complex problem-solving and planning, offer subpar writing suggestions to end-users, and generate biased and stereotypical simulations of various socio-demographics for scientific research.
> >
> >
> > ---
> > **Question: Can you explain the choice of personas in the Race and Gender Categories?**
> >
> > We chose a few representative samples within these categories to get a picture of the extent of bias in these models within a reasonable budget. We will clarify in the paper that by “Race”, we generally refer to Race and Ethnicity (African would also fall under the ethnicity category as it is used in countries such as the UK). We didn’t use African American to reduce the bias towards American definitions of race/ethnicities.
> >
> > Since our coverage of gender was especially biased towards the dominant categories, we have also extended our study to include trans + non-binary genders (please refer to the general response for the corresponding results and analysis).

---

### Official Review · Reviewer_34gE · 2023-10-30

**Soundness:** 2 fair
**Presentation:** 2 fair
**Contribution:** 2 fair
**Rating:** 5
**Confidence:** 4

**Summary:**

The paper studies how personas can affect reasoning capabilities of language models and biases that can arise as a result of this. Authors perform different empirical evaluations on various personas and datasets and showcase the existing disparities amongst different groups/personas. Authors also show that prompt based bias mitigation approaches have minimal effect in reducing biases studied in this work.

**Strengths:**

1. The paper is written clearly and is easy to follow.

2. The paper studies a timely and important issue.

3. Different datasets are studied in this work.

**Weaknesses:**

1. The experimental results are shallow and are not discussed in detail. The results only touch the surface area of what authors have observed. I would have loved to see more in depth discussion on what was the reason behind these observations.

2. Only a simple prompt base mitigation strategy was considered which was not that effective. It would have been good if authors have considered more sophisticated approaches to address this issue.

3. The studied personas were limited and authors did not consider intersectional biases.

4. Only gpt-3.5-turbo was used in this study. It would have been interesting to see how different models perform differently in terms of persona biases.

Overall, since the paper was mostly empirical, I would have loved to see more detailed studies and ablations with a larger pool of models and more in depth discussion on the findings and results to fully support the claims.

**Questions:**

I am still not 100% clear and convinced on why demographics, such as religious, should perform lower than each of the more fine-grained demographics, such as Atheist, Christian, and Jewish.

---

> ### Author Response · Authors · 2023-11-23
> **Response from the authors to Reviewer 34gE**
>
> Thank you for recognizing the importance of this work and sharing your valuable feedback! Please see our responses below to your specific questions and concerns.
>
> ---
> **Would different models perform differently?**
>
> We have significantly expanded our analysis to include Llama2-70b-chat, GPT-4-Turbo, and GPT3.5-Turbo Nov. We find that these models do perform differently (compared to ChatGPT) both in the extent and pattern of the bias. For instance, we observe that the latest version of GPT3.5-Turbo exhibits significantly higher levels of bias, whereas GPT-4-Turbo has reduced levels of bias across personas. Additionally, we see a difference in the nature of the bias as well. For instance, compared to ChatGPT, there is a significant reduction in the amount of bias against Physically Disabled in Llama-2, whereas the amount of bias against the non-binary genders goes up in Llama-2.
>
> ---
> **Non-binary genders & Intersectional biases**
>
> Thanks for the suggestion to extend the scope of our study. We have added results for *3 non-binary gender personas and 13 compound personas*. With these additions, our bias study now includes insights from 1.5 Million model predictions (more than 200 Million generated tokens) and spans 32 socio-demographic personas, 24 datasets, and 4 LLMs.
>
> We found evidence of bias against non-binary genders across models (Figs. 5, 12, 16, 18). We additionally found that intersecting 2 personas with high levels of bias can at times reduce the overall amount of bias, whereas intersecting personas with high and low biases usually results in a level of bias that is in between the bias levels of the intersecting personas. Please refer to the general response for additional findings. The results for the non-binary genders are included in the main body of the updated paper (Figs. 2, 3, 4, 5), and the results and analysis for the compound personas are in the newly added Appendix E.
>
> ---
> **More sophisticated approaches to address the bias issue?**
>
> Thanks for the suggestion. We have extended our set of debiasing prompts and have added results for 8 additional prompts (see the general response above). Our explored prompts are stylistically and semantically diverse that range from providing a nudge (“Try your best”) to strongly instructing to answer a question (“Don’t refuse”) to even making each persona a subject expert. In total, we tried 12 de-biasing prompts exploring this space and did not see any notable differences. Note that the focus of our work is to identify and characterize the reasoning biases in persona-assigned LLMs, and to the best of our knowledge, ours is the first work to do so. We leave more sophisticated bias mitigation approaches such as aligning LLMs on our dataset of 1.5 Million released model outputs to future work.
>
> ---
> **More in-depth discussion; reason behind these observations.**
>
> Section 4 of our paper analyzed the model behavior and identified two key manifestations of the biased reasoning: explicit abstentions and implicit errors. We showed in Sec. 4 that while abstentions form a large percentage of the model errors (Fig. 8), the bias goes beyond abstentions and implicitly affects the model’s reasoning, leading it to make more mistakes for certain socio-demographics (Fig. 9).
>
> For a more causal understanding of these observations, we can consider the model development process. Given the key sources of data in these models are the pre-training corpus, instruction-tuning datasets and the RLHF training data, we believe these are the key reasons for this observed bias. Biases within the datasets used at each stage can favor or disfavor certain socio-demographic groups (e.g. text snippets like ‘religious individuals lack interest in sciences,’ or ‘Jewish individuals possess high intelligence’), consequently shaping the induced biases in these models.
>
> Given the access to these datasets (e.g. C4), one could possibly run ablations to identify the more specific cause. E.g. If pre-training on a corpus excluding certain subreddits results in reducing the bias, we can identify that subreddit as a potential reason behind these observations. However, such ablations would be pretty costly and not even feasible with GPT models where the entire model building process is private.  We will add this to the discussion section in the final version of the paper.
>
> ---
> **Question: Why does the religious persona perform lower than each of the more fine-grained demographics?**
>
> It is hard to say for certain. The model’s opinions about a persona are not guaranteed to be consistent with the hypernym-homonym relations. We theorize that models pick up certain incorrect stereotypes about the religious people from the pre-training corpus (e.g. “religious people are not open-minded”, “religious people don’t accept sciences”) that are probably not that commonly expressed in written text for more fine-grained religions.

---

### Official Review · Reviewer_jUKs · 2023-11-06

**Soundness:** 3 good
**Presentation:** 4 excellent
**Contribution:** 3 good
**Rating:** 8
**Confidence:** 5

**Summary:**

The authors study how specifying different personas (based on political ideology, religious beliefs, race, gender, etc) in the initial prompts for ChatGPT (gpt-3.5-turbo specifically) results in worse performance on a slew of reasoning task evaluations. It is noted the when specifically asked whether these personas would impact the ability to answer reasoning task questions the LLM responds that the type of persona will not matter. Unfortunately, as the authors discover, the bias is baked deep into the model, and that protections are merely superficial. The authors created a "human" and "average human" baseline persona to compare against.

The "physically disabled" persona does much worse on the evaluations reasoning tasks compared to the human persona. Of note, binary gender personas do not impact the reasoning capabilities.

**Strengths:**

The paper presents an interesting question and provides solid results. The paper is by-and-large well written. The paper provides enough detail such that the results can be reproduced and built upon. The paper does a good job citing the relevant literature,

**Weaknesses:**

From a writing standpoint, I wish the authors did not abbreviate "statistically significant", and given that they chose to do so they should be consistent with the abbreviation. The authors do not specify explicitly what the "human" and "average human" persona prompts are. I assume the term "human" was put in place of the {persona} holder showed in Table 1, but I wish that I hadn't needed to assume. I also wish they had provided results for these datasets when no persona was provided. I wish the authors had performed similar analysis on a different LLM; I am left unsure if this is an issue exclusively for gpt-3.5 or is it in all LLMs.

**Questions:**

- In section 5 the authors note "it relies on the task requiring a well-defined expertise that can be succinctly specified in the prompt".  Would it be possible in the prompt just to specify the LLM should as an "expert in the subject" or "domain expert". Why the need to explicitly state the LLM should adopt the persona of "chemist" or "lawyer".
- How does the LLM perform on these tasks when no persona is provided?
- In table 8 which of the persona prompts succeeded and which failed your test?
- Did you consider how intersectionality could alter the results (i.e. you are a Hispanic life-long Republican). Would be really interesting.

---

> ### Author Response · Authors · 2023-11-23
> **Response from the authors to reviewer jUKs**
>
> Thank you for taking the time to review our paper! We are glad to hear that you found our study intriguing and our findings robust. Please see our responses to your questions and concerns below.
>
> ---
> **Analyses on other LLMs**
>
> Thanks for this suggestion. We have expanded our analysis to Llama2-70b-chat, GPT-4-Turbo, and GPT3.5-Turbo Nov. We observed that the persona-induced biases are not exclusive to GPT-3.5 and even extend beyond the OpenAI family of models. We additionally found the extent and nature of the bias to vary across the models. Please refer to the general response and the newly added Appendix D for additional results and findings.
>
> ---
> **How does intersectionality alter the results?**
>
> Thanks for this interesting question. We have added a systematic analysis of 13 compound personas that intersect the existing socio-demographic personas. We found that intersecting a low bias persona (e.g. ‘a man’) with a high bias persona (e.g. ‘a physically-disabled person’) does mitigate some of the bias in the high bias persona but not completely. Interestingly, we found that intersecting two low bias personas (e.g. ‘a physically-disabled person’ and ‘a religious person’) can at times achieve a lower overall bias compared to the participating personas. Please refer to the general response and the newly added Appendix E for more details.
>
> ---
> **Why not just prompt for "expert in the subject" or "domain expert”?, Why the need to explicitly state “Chemist”?**
>
> This is a very interesting suggestion. We experimented with 3 new (*’domain expert’*) prompts based on this suggestion and found them to have little to no effect. We also experimented with 5 other stylistically different debiasing prompts (see the general response above) and found them to have little effect as well (more details in Appendix C).
>
> It's interesting that a generic instruction of “act like a domain expert” does not work but grounding the expertise to a task-specific occupation (e.g. Chemist) works so well. It is possible that the domain expert terminology still leaves room for the LLM to surface its biases (e.g., by assuming that different personas have different levels of domain expertise), whereas specifying an occupation potentially unifies the perceived capabilities across personas.
>
> ---
> **How does the LLM perform when no persona is provided?**
>
> This is a good question. We evaluated ChatGPT with a no persona prompt (i.e. only providing the task instruction) and found that it basically performs the same as the “Human” persona. E.g. With no persona, ChatGPT achieves an aggregate score of *0.66* on our 24 dataset benchmark as compared to the *0.65* score for the human persona (a non statistically significant difference). We also didn’t observe any statistically significant change on any of the datasets between these two personas. Thus, all of our bias findings hold relative to this ‘no persona’ baseline as well.
>
> ---
> **Be consistent with the abbreviation.**
>
> Thank you for pointing this out. We have fixed this and use a consistent abbreviation in the updated version of the paper.
>
> ---
> **Specify explicitly what the "human" and "average human" persona prompts are.**
>
> You are correct -- these are generated by replacing the {persona} placeholders in our instruction templates. We have clarified this in Section 3.1 of the updated paper (see changes in blue).

---

### Author Response · Authors · 2023-11-23
**General response from the authors to all reviewers**

We would like to thank all the reviewers for taking the time to review our work and share their valuable feedback. We thank the reviewers for recognizing the timeliness and importance of this work (Rev. 34gE, s6hK) as well as appreciating the empirical analysis (Rev. jUKs, mMGg). In response to reviewers' helpful suggestions, we have significantly extended the scope of our study and have added results and analysis for 16 additional personas, 3 more LLMs, and 8 new mitigation instructions. With the addition of these new results, our study now represents findings from over 1.5 Million model predictions spanning 24 datasets, 32 socio-demographic personas, and 4 LLMs. Given the immense commercial excitement and momentum around personified LLMs ([character.ai](https://beta.character.ai/), [Meta AI Chatbots](https://ai.meta.com/genai/), [Replika](https://replika.com/), [OpenAI Custom GPTs](https://openai.com/blog/introducing-gpts)), it is critical to understand how persona-assigned LLMs can adversely affect real people and we believe our study takes an important step in this direction. We will release all our outputs and believe this large-scale dataset of model predictions with the corresponding rationales will enable further analyses and mitigation of persona-induced biases in these models.


We have addressed each reviewer's comments individually and below we present the key findings from the new experiments that cater to the requests from multiple reviewers. We have added **4 new sections** to the Appendix of the revised version of the paper to share our findings in detail (Appendix D, E, F and G). To make our changes easy to find, we have marked them in **blue** (new images and tables are marked with blue captions and new sections with blue section headers) and have added **[NEW]** and **[UPDATED]** tags to the section headers to denote new and updated sections respectively.

---
## New Experiments and Results

- **Other LLMs**: We have extended our analysis to *3 other LLMs*: LLama2-70B-chat, and 2 latest OpenAI models – GPT-4-Turbo, and GPT3.5-Turbo Nov (gpt-3.5-turbo-1106) – as per the request from reviewers jUKs, 34gE, and s6hK. We find that persona-induced biases are also prevalent in these models and its extent (GPT3.5-Turbo Nov is more biased than GPT3.5-Turbo June; GPT-4-Turbo less so) and nature (e.g. higher gender bias in Llama2 model) varies across models. For more detailed results, please see **Appendix D**.

&nbsp;
- **Intersectional Biases**: We have added results and analysis for *13 new compound personas* (suggested by reviewers jUKs, 34gE) at the intersection of our existing 16 socio-demographic personas, e.g. *“a physically-disabled Religious person”*. We observe that such intersecting personas have scores that either lie between the two input personas’ scores or are higher than both of them (depending on the extent of the bias in the intersecting personas). We share more detailed results in **Appendix E**.

&nbsp;
- **More De-biasing experiments**: We have expanded our bias mitigation experiments to include *8 additional de-biasing prompts* (suggested by jUKs, 34gE, and s6hK). These include variations of the existing de-biasing prompt styles as well as new prompt styles such as *“you are a domain expert”*(suggested by jUKs). Consistent with our findings before, we find that none of these de-biasing prompts are effective at mitigating the bias. We share the prompts and discuss these results in the updated **Appendix C**.

&nbsp;
- **Beyond binary gender**: As per reviewer mMGg’s suggestion, we have added results for 3 new  gender personas: 'a non-binary individual', 'a transgender man,' and 'a transgender woman' and have updated the figures in the main body of the paper to include these results (see **updated Figs. 2-5**). Compared to the lack of bias against binary genders, we did notice bias against these “non-traditional” gender personas (we’ll update the final version of the paper to surface these new findings).

---

### Meta-Review · Area_Chair_TvMY · 2023-12-03

**Metareview:**

This paper investigates the effect several persona assignments on ChatGPT’s ability to perform basic reasoning tasks, which span 24 datasets. The authors find large drops in accuracy for certain personas (including physically disabled individuals and religious individuals). The reviewers have mixed opinion though the main argument of the reject rating is that only ChatGPT was investigated. This lead to a revision and the paper now covers several large language models. They all show persona-induced biases. While indeed, there could be done much more such as investigating more mitigation strategies, additional forms of biases, I do expect a lot of future research motivated by the present work. Hence I suggest to accept paper, also because all reviewers agree that the findings are interesting and important.

**Justification For Why Not Higher Score:**

We know that LLMs have biases and capture several stereotypes. The present paper however shows that they are probably much deeper.

**Justification For Why Not Lower Score:**

There will be many more LMM-based personas in the future. Showing that stereotypes can be so easily also enter the "reasoning" of LLMs by just personas is important to know

---

### Decision · Program_Chairs · 2024-01-16

Accept (poster)